# Effect of Safinamide on Non-Motor Symptoms and Quality of Life in Parkinson’s Disease Patients According to Sex, Age, Disease Duration and Levodopa Equivalent Daily Dose

**DOI:** 10.3390/brainsci15070666

**Published:** 2025-06-20

**Authors:** Ángela Solleiro Vidal, Rosa Yáñez Baña, Carmen Labandeira Guerra, Maria Icíar Cimas Hernando, Iria Cabo López, Jose Manuel Paz González, Maria Gema Alonso Losada, Diego Santos García

**Affiliations:** 1Grupo de Investigación en Enfermedad de Parkinson y Otros Trastornos del Movimiento, INIBIC (Instituto de Investigación Biomédica de A Coruña), 15006 A Coruña, Spain; 2Complejo Hospitalario Universitario de Ourense (CHUO), 32005 Ourense, Spain; rosa.maria.yanez.bana@sergas.es; 3Complejo Hospitalario Universitario de Vigo (CHUVI), 36312 Vigo, Spain; carmen.labandeira@hotmail.com (C.L.G.); gemavarita@gmail.com (M.G.A.L.); 4Hospital de Povisa, 36211 Vigo, Spain; icimash@gmail.com; 5Complejo Hospitalario Universitario de Pontevedra (CHOP), 36071 Pontevedra, Spain; icabol@yahoo.es; 6Department of Neurology, Hospital Universitario de A Coruña (HUAC), Complejo Hospitalario Universitario de A Coruña, 15006 A Coruña, Spain; jpazg1@hotmail.com; 7Hospital San Rafael, 15006 A Coruña, Spain; 8Fundación Degen, 15004 A Coruña, Spain

**Keywords:** non-motor symptoms, post hoc analysis, Parkinson’s disease, quality of life, safinamide, sex, disease duration

## Abstract

**Background and objective:** Safinamide can improve the non-motor symptoms (NMSs) and quality of life (QoL) in patients with Parkinson’s disease (PD). In this post hoc analysis of the SAFINONMOTOR study, we analyzed the effect of safinamide on NMSs and QoL according to age, sex, disease duration (DD), and levodopa equivalent daily dose (LEDD). **Patients and Methods:** The change from baseline to the end of the observational period (6 months) in the Non-Motor Symptoms Scale (NMSS) and the 39-item Parkinson’s Disease Quality of Life Questionnaire (PDQ-39) was analyzed in subgroups according to sex (male vs. female), age (≤75 vs. >75 years old), DD (≤10 vs. >10 years) and LEDD (≤1000 vs. >1000 mg). Nonparametric tests and general linear model (GLM) repeated measures were applied. **Results:** A total of 44 patients completed the final visit and were valid for the analysis. A significant reduction in the NMSS score was observed in all groups. Regarding QoL, a significant reduction in the PDQ-39 score was observed in females (*p* < 0.0001) and in patients with a DD > 10 years (*p* = 0.011) but not in males or those > 75 years old or receiving an LEDD > 1.000 mg. In the GLM, only LEDD at baseline influenced the degree of change in the NMSS total score (*p* = 0.026; F = 5.23). None of the variables influenced the change in the PDQ39. **Conclusions:** Safinamide improved NMSs independently of sex, age, DD, and LEDD. QoL improved independently of DD, and in females and non-elderly and very treated patients.

## 1. Introduction

Parkinson’s disease (PD) is the second most common neurodegenerative disorder, following Alzheimer’s disease [1], and is characterized by the degenerative loss of dopaminergic neurons in the substantia nigra [2]. While it is primarily distinguished by motor symptoms, PD patients may also present a wide range of non-motor symptoms (NMSs) such as depression, cognitive impairment, anxiety, sleep disorders, and pain [3,4], all of which can significantly reduce their quality of life (QoL) [5,6]. The primary treatment for PD is dopamine replacement therapy, with levodopa as the gold standard for managing motor symptoms [7]. However, prolonged use is associated with motor complications such as the “wearing-off” phenomenon and the development of dyskinesia. For this reason, as the disease progresses, the use of adjuvant therapies and other antiparkinsonian drugs becomes necessary to complement levodopa [8]. Although current therapies have successfully addressed the majority of motor features, NMSs do not improve always with dopamine replacement, making their treatment a challenge. This therapeutic need has driven research into drugs that modulate the activity of neurotransmitter systems other than dopamine, with the goal of alleviating these symptoms in PD [9,10].

In this context, safinamide, an oral α-aminoamide derivative [11], has been shown to be an effective adjunct therapy option to levodopa [12]. It has been approved in various regions, including Europe, the United States, Japan, and other parts of Asia, for the treatment of motor fluctuations in PD [13,14]. Specifically, safinamide has demonstrated benefits for motor symptoms such as tremor, bradykinesia, rigidity, and gait [15,16]. The distinctive mechanism of action of safinamide affects the dopaminergic pathways through the selective and reversible inhibition of monoamine oxidase type B (MAO-B), and glutamatergic pathways through the blocking of sodium channels and modulation of calcium channels [17]. Beyond its effects on motor function, recent studies have shown that safinamide also has a positive impact on NMSs and QoL [18,19,20]. Specifically, we observed in the SAFINONMOTOR study a positive impact of safinamide on NMSs and QoL in 50 PD patients who had a severe or very severe NMS burden after 6-month follow-up [21]. Moreover, benefits were observed in mood, sleep, daytime sleepiness, as well as pain in other analyses from this study [22,23,24]. Despite these advances, important questions remain about the extent and mechanisms of safinamide’s action, particularly concerning its role in alleviating NMSs [18]. Factors such as sex, age, disease duration (DD), and daily levodopa dose could influence its effectiveness, making it crucial to understand these interactions in order to optimize personalized treatment for the disease. However, to date, no specific studies have addressed the effect regarding all these variables.

Therefore, the objective of this post hoc analysis of the SAFINONMOTOR study was to evaluate the effect of safinamide on NMS burden and QoL with respect to sex, age, DD, and daily levodopa equivalent dose (LEDD) in PD patients, in order to provide further evidence of its clinical impact and potential applications in personalized treatment for PD.

## 2. Materials and Methods

SAFINONMOTOR (an open-label study of the effectiveness of SAFInamide on NON-MOTOR symptoms in Parkinson’s disease patients) is a prospective, observational (Phase IV), open-label, multicenter, single-nation (Spain), and follow-up study conducted in 5 centers from Spain [21]. Patients with PD according to the United Kingdom Parkinson’s Disease Society Brain Bank criteria [25] without dementia [26] who were considered for treating with safinamide by the neurologist having a NMSS total score ≥ 40 were included. Methods from the SAFINONMOTOR are available in: https://www.ncbi.nlm.nih.gov/pmc/articles/PMC7999475/ (accessed on 22 May 2025) [21].

This study included 4 scheduled visits: V0 (baseline; before starting safinamide); V1M (1 month ± 7 days); V3M (3 months ± 15 days); V6M (6 months ± 15 days, end of the observation period). With the aim of assessing NMSs and PD-related QoL, patients completed both the Non-Motor Symptoms Scale (NMSS) [27] and the 39-item Parkinson’s disease Questionnaire (PDQ-39) [28] in all visits. The NMSS includes 30 items, each with a different non-motor symptom. The symptoms refer to the 4 weeks prior to assessment. The total score for each item is the result of multiplying the frequency (0, never; 1, rarely; 2, often; 3, frequent; 4, very often) x severity (1, mild; 2, moderate; 3, severe) and will vary from 0 to 12 points. The scale score ranges from 0 to 360 points (NMSS total score). The items are grouped into 9 different domains: (1) cardiovascular (items 1 and 2; score range, 0 to 24); (2) sleep and fatigue (items 3 to 6; score range 0 to 48); (3) depression and apathy (items 7 to 12; score range 0 to 72); (4) perceptual problems and hallucinations (items 13 to 15; score range 0 to 36); (5) attention and memory (items 16 to 18; score range 0 to 36); (6) gastrointestinal tract symptoms (items 19 to 21; score range 0 to 36); (7) urinary symptoms (items 22 to 24; score range 0 to 36); (8) sexual dysfunction (items 25 and 26; score range 0 to 24); and (9) miscellaneous symptoms (items 27 to 30; score range 0 to 48). The PDQ-39 is a Parkinson’s disease-specific questionnaire designed to assess patients’ quality of life. It comprises 39 items distributed across 8 domains: (1) mobility (items 1–10); (2) activities of daily living (items 11–16); (3) emotional well-being (items 17–22); (4) stigma (items 23–26); (5) social support (items 27–29); (6) cognition (items 30–33); (7) communication (items 34–36); and (8) pain and discomfort (items 37–39). Each item is scored from 0 (never) to 4 (always), referring to symptoms experienced during the 4 weeks prior to assessment. Sociodemographic data, PD-related factors, comorbidities, and treatments were also collected. Moreover, other scales were administered by protocol in different visits of this study.

In relation to the purpose of this analysis, subgroups were defined according to sex (male vs. female), age (>75 vs. ≤75 years), disease duration (DD) (>10 vs. ≤10 years), and LEDD (>1000 vs. ≤1000 mg). The cut-off points established for the 3 groups with numerical variables were based on the literature and the possible implications from the point of view of daily clinical practice [29,30,31].

Safinamide was administered as a 50 mg tablet once daily for one month and increased to 100 mg/day at V2. In some cases (e.g., dyskinesia), the 100 mg dose could be introduced earlier, or the dose could be maintained at 50 mg/day based on the neurologist’s judgment. No medication other than safinamide could be modified (schedule, dosage, etc.) during the follow-up unless deemed absolutely necessary by the neurologist. All changes were recorded, including medications related to Parkinson’s disease and non-related medications, as well as the LEDD [32].

The safety dataset included all subjects who initiated the study device. Safety analyses were assessed by adverse events (AEs). All AEs were coded using the current version of the Medical Dictionary for Regulatory Activities (MedDRA). The number and percentage of subjects with treatment-emergent AEs were provided according to the MedDRA system organ class and preferred term, classified by severity and relationship to the study treatment as evaluated by the investigator for all subjects.

### 2.1. Data Analysis

Data were analyzed using SPSS version 20.0 for Windows. Continuous variables were expressed as mean ± standard deviation (SD) or median and quartiles, depending on data distribution. Normality was checked using the Kolmogorov–Smirnov test for a single sample.

Only patients who completed the final visit at 6 months (V6M) were valid for this post hoc analysis. Each domain of the NMSS and PDQ-39 was expressed as a percentage: (score/total score) × 100. A Wilcoxon rank-sum test was performed to assess the change in the total score and in each domain score of the NMSS and PDQ-39 from baseline visit (V0) to the end of the observation period at six months (V6M) in each previously defined subgroup: males; females; ≤75 years old; >75 years old; ≤10 years of DD; >10 years of DD; ≤1000 mg of LEDD; >1000 mg of LEDD. Cohen’s d formula was applied for measuring the effect size, which was considered to be absent, <0.2; small, 0.2–<0.5; moderate, 0.5–<0.8; large, 0.8–1.3; or very large, ≥1.3. Additionally, a general linear model for repeated measures (GLM) was used to analyze the impact of the variable under study (i.e., sex, age, DD, and LEDD) on the change experimented by the patients in the total score of the NMSS and PDQ-39 from V0 to V6M. Differences in the score of the NMSS and PDQ-39 between patients from a different subgroup according to the same variable (i.e., male vs. female, etc.) at V0 and V6M were analyzed using the Mann–Whitney U test. The frequency of patients with any adverse event was compared between subgroups using the Chi-squared test (all sample). Values of *p* < 0.05 were considered statistically significant.

### 2.2. Standard Protocol Approvals, Registrations, and Patient Consents

This study received approval from the Clinical Research Ethics Committee of Galicia (2018-052; 28/FEB/2019). Written informed consent was obtained from all participants prior to study initiation. SAFINONMOTOR was classified by the AEMPS (Spanish Agency for Medicines and Health Products) as a post-authorization prospective follow-up study with the code DSG-SAF-2018-01.

## 3. Results

Fifty PD patients were included in the SAFINONMOTOR study between May 2019 and February 2020 (age 68.5 ± 9.1 years; 58% females). Data about sociodemographic aspects, comorbidities, antiparkinsonian drugs, and other therapies were previously published [21]. Forty-four PD patients (88%; age 69.1 ± 9.3 years; 54.5% females) completed the final visit (V6M) and were valid for this post hoc analysis.

A significant reduction in the NMSS total score was observed in all subgroups, including specifically patients > 75 years old (*p* = 0.040), with a DD > 10 years (*p* = 0.015), and those receiving an LEDD > 1000 mg (*p* = 0.002) (Table 1 and Figure 1). By domains, a significant decrease in the score from V0 to V6M was detected in “Mood/apathy” and “Miscellaneous” in all subgroups (Table 2). At the other extreme, no significant change was found in “Cardiovascular”, “Perceptual symptoms” or “Sexual dysfunction” in any of the subgroups (Table 2). “Sleep/fatigue”, “Attention/memory”, and “Gastrointestinal symptoms” improved in males whereas “Urinary symptoms” in females (Table 2). Moreover, “Urinary symptoms” improved independently of age and LEDD (i.e., in both subgroups defined of the variable) (Table 2). Regarding QoL, a significant reduction in the PDQ-39 score was observed in females (*p* < 0.0001) and in patients with a DD > 10 years (*p* = 0.011) but not in males or those > 75 years old or receiving an LEDD > 1000 mg (Table 3 and Figure 2). By domains, the “Emotional well-being” score decreased significantly in all defined subgroups except patients older than 75 years old (Table 4). Improvement in other domains was not detected in patients older than 75 years old, with a DD longer than years, or an LEDD higher than 1000 mg (Table 4). In the GLM, only the LEDD at baseline influenced the degree of change in the NMSS total score from V0 to V6M (*p* = 0.026; F = 5.23), but the result was not significant (*p* > 0.05) after adjustment to covariates (sex; age; disease duration). The decrease in the NMSS total score was 43.8% for the group with an LEDD > 1.000 mg/day vs. 33.7% for those ≤ 1.000 mg. None of the variables influenced the change in the PDQ39.

As previously published [21], a total of 21 adverse events were reported in 16 patients (32%), including 5 classified as severe, though none were attributed to safinamide. The most frequent reported (6%) were dyskinesias and nausea. Six participants withdrew from this study for the following reasons: one due to withdrawal of consent; one due to discontinuation of safinamide following a deep brain stimulation procedure (recorded as a SAE due to hospitalization); one due to a personal decision citing lack of efficacy; and three due to adverse events (two cases of dizziness and one respiratory infection). Only one patient discontinued treatment due to an adverse event considered related to safinamide (dizziness). By groups, no differences were detected in the frequency of developing at least one AE: males 30% vs. females 20.8% (*p* = 0.362); ≤75 years old 24.2% vs. >75 years old 27.3% (*p* = 0.565); DD ≤ 10 years 20.6% vs. DD > 10 years 40% (*p* = 0.200); LEDD ≤ 1000 mg 16.7% vs. LEDD > 1000 mg 43.9% (*p* = 0.070).

## 4. Discussion

In the previously published SAFINONMOTOR study [21], we observed improvement in the NMSs and QoL among patients with PD after six months of treatment with safinamide. In this post hoc analysis, we found that the significant improvement detected in the NMS burden was present across all groups, including in those patients with more clinically complex profiles, such as patients over 75 years old, those with a disease exceeding 10 years, and patients receiving a high LEDD (>1000 mg/day). As for QoL, gains were noted across the board but reached statistical significance particularly in women, patients no older than 75 years old, and those receiving an LEDD no higher than 1000 mg, regardless of disease duration (≤10 years or >10 years).

These findings align with the previous research supporting the efficacy and/or effectiveness of safinamide in reducing the overall NMS burden [20,33,34,35]. Importantly, this is the first study to demonstrate a significant NMS improvement across diverse clinical profiles, regardless of sex, age, DD, or LEDD. These results highlight the potential of safinamide as an adjunct therapy for NMSs across various patient subgroups and stages of progression. The observed benefit is particularly striking in individuals with more advanced conditions—those over 75 years old, with more than 10 years of disease progression, or requiring higher doses of levodopa—who experience a higher symptom burden and poorer QoL. This subgroup is becoming increasingly common in routine clinical practice, further underscoring the clinical value of these results. In Japan, for instance, two-thirds of patients with PD treated with safinamide from the J-SILVER study [36] were over 75 years old, emphasizing the importance of having effective and well-tolerated treatments for this population. Safinamide improved motor and NMSs and QoL in older patients with PD in the early stages of wearing-off in the mentioned J-SILVER study [36]. Data from other studies also demonstrated the effectiveness of safinamide on motor and NMSs in elderly and/or more advanced PD patients [12,37,38,39].

More specifically, a significant improvement has been consistently reported in domains such as mood, sleep, urinary symptoms, and pain [40,41,42]. Here, the strongest effect of safinamide was observed on mood and miscellaneous, which includes pain, that was detected in all groups independently of the sex, age, DD, and LEDD. Previous studies have reported the differences in PD patients in their clinical profile according to sex [43], suggesting for our findings that a different response to safinamide in some NMSs could be expected. For example, we found a better response for urinary symptoms in females, something important but not analyzed in previous reports showing improvement in urinary symptoms as a whole [44,45]. Data from pivotal studies indicated that safinamide has positive effects on both motor and non-motor functions in patients with Parkinson’s disease, with comparable efficacy observed in both genders [15,16,17,46]. Regarding pain, one of the most common and limiting NMSs in PD [47], the results of this study are particularly relevant. A significant improvement was observed in pain-related symptoms collected with the NMSS across all subgroups analyzed, independent of demographic or clinical characteristics such as sex, age, levodopa dose, or disease duration, further underscoring the therapeutic potential of safinamide in this area. With prevalence estimates ranging from 30 to 50%—and up to 85% when accounting for all pain types [48]—its effective management remains a pressing clinical need. Alarmingly, despite its high prevalence and impact on QoL, nearly half of patients suffering from pain do not receive any analgesic intervention, a number that could rise to 3.7 million globally by 2030 [48]. This treatment gap stands in stark contrast with the growing evidence supporting the use of safinamide as an adjunctive strategy. Indeed, prior investigations, such as those by Geroin et al. [42], have reported significant improvements in the KPPS, BPI, and NRS scales after three months of treatment, while post hoc analyses [49] and systematic reviews with meta-analyses [35] have consistently supported its efficacy in relieving pain within the PD population. In this context, since the majority of the patients in the SAFINONMOTOR study were receiving a dose of 100 mg per day at the end of this study, the influence of the safinamide on the glutamatergic pathways cannot be ruled out, as other authors have previously suggested [50,51].

Regarding QoL, safinamide has demonstrated beneficial effects on patients’ QoL [19,21,52,53]. Here, although the improvement was widespread, it was only significant in women, patients ≤75 years old, and those receiving ≤1000 mg of levodopa, regardless of how long patients had been living with the disease (≤10 years or >10 years). Notably, the improvement in QoL observed among women is especially meaningful, given their increased vulnerability and higher likelihood of experiencing depressive symptoms [23,54,55]. This subgroup exhibited a significant reduction in emotional well-being scores on the PDQ-39 scale. In a post hoc analysis of the XINDI study, Pellecchia et al. [56] observed greater improvements in females compared to males in the social and emotional domains such as activities of daily living, bodily discomfort, and emotional well-being. In contrast, we did not find significant reductions in men or in patients over 75 years old. This may reflect group-based differences in therapeutic response or QoL perception, though it could also stem from limited statistical power due to small sample sizes. A similar pattern was seen in patients receiving more than 1000 mg of LEDD, where improvements did not reach statistical significance. Nonetheless, the previous studies, including that of Peña et al. [40], have highlighted safinamide’s positive effect on mood, with consistent gains in the PDQ-39 emotional well-being domain in both short- and long-term analyses [19], reinforcing its role as a valuable adjunctive treatment for NMSs in PD.

The present study has some important limitations. First, the open-label design, which lacked a placebo group, introduces a potential risk of bias due to treatment expectancy. However, the observed improvements and their stability over time support the clinical relevance of the results. Second, the sample size was relatively small, particularly in certain subgroups, which may have limited the statistical power to detect significant differences. Third, the inclusion of patients with a severe or very severe NMS burden may reduce the generalizability of the results to all patients with PD. Fourth, the presence of antidepressant treatment in some participants, which could have influenced the interpretation of safinamide’s effects on mood. However, the favorable pharmacological profile of safinamide minimizes the risk of significant drug interactions. Finally, the COVID-19 pandemic impacted data collection, with four follow-up visits conducted via telephone. This shift in assessment method may have influenced the subjective evaluations of mood and QoL.

Overall, our results demonstrate that safinamide treatment over six months significantly improved NMSs in PD patients across all analyzed subgroups, irrespective of sex, age, disease duration, or levodopa dose, reinforcing its clinical utility as an adjunctive therapy in PD. Concerning QoL, an overall improvement was also observed after six months of treatment, with significant gains particularly in women, patients aged ≤75 years old, and those receiving an LEDD ≤ 1000 mg. These findings emphasize the importance of considering a patient’s individual clinical profile when making therapeutic decisions and highlight safinamide’s value as an effective tool for enhancing the comprehensive management of PD. Further studies with larger sample sizes are needed to validate these results and explore more thoroughly how treatment response varies across demographic and clinical factors in order to optimize safinamide’s clinical use and maximize its therapeutic benefits for non-motor symptoms in PD patients.

## Figures and Tables

**Figure 1 brainsci-15-00666-f001:**
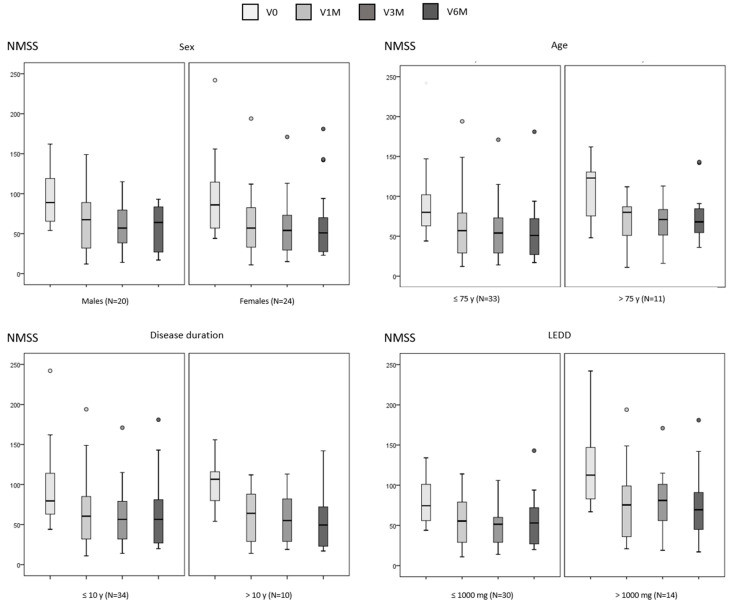
Change in the NMSS total score from baseline (V0) to the final visit at 6 months (V6M) in PD patients from the SAFINOMOTOR study according to sex, age, disease duration, and LEDD (data of all visits are shown). Data are presented as box plots, with the box representing the median and the two middle quartiles (25–75%). Mild outliers (O) are data points that are more extreme than Q1—1.5 (case number shown). LEDD, levodopa equivalent daily dose; NMSS, Non-motor Symptoms Scale.

**Figure 2 brainsci-15-00666-f002:**
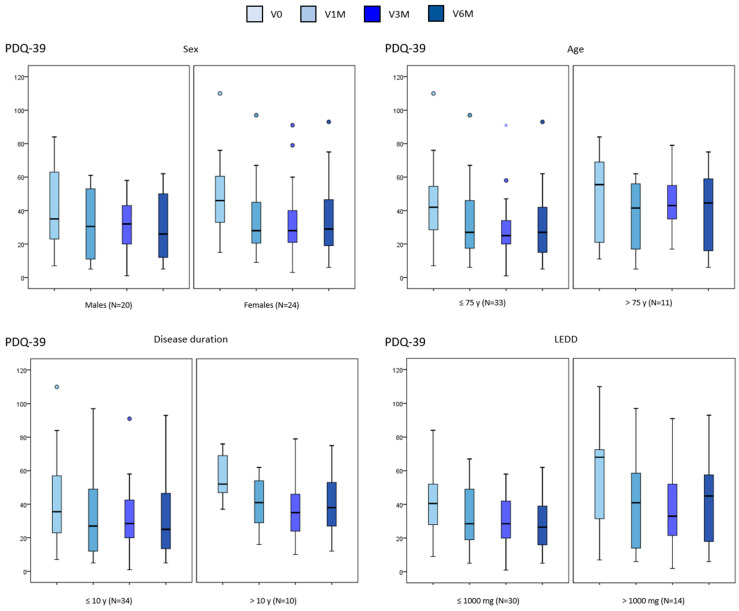
Change in the PDQ39 total score from baseline (V0) to the final visit at 6 months (V6M) in PD patients from the SAFINOMOTOR study according to sex, age, disease duration, and LEDD (data of all visits are shown). Data are presented as box plots, with the box representing the median and the two middle quartiles (25–75%). Mild outliers (O) are data points that are more extreme than Q1—1.5 (case number shown). LEDD, levodopa equivalent daily dose; NMSS, PDQ-39, 39-item Parkinson’s Disease Quality of Life Questionnaire.

**Table 1 brainsci-15-00666-t001:** Change in the NMSS total score from baseline (V0) to the final visit at 6 months (V6M) in PD patients from the SAFINOMOTOR study according to sex, age, disease duration, and LEDD.

	At V0	At V6M	Cohen’s d	*p* ^a^	*p* ^b^	*p* ^c^
Sex					0.629	0.494
Males (N = 20)	93.6 ± 32.4	59.4 ± 27.7	−1.8	**<0.0001**		
Females (N = 24)	93.6 ± 45.2	60.4 ± 41.5	−1.3	**0.001**		
Age					0.069	0.050
≤75 y (N = 33)	88.6 ± 38.9	54.2 ± 34	−1.9	**<0.0001**		
>75 y (N = 11)	108.7 ± 38.6	77.2 ± 35.6	−1.0	**0.040**		
Disease duration					0.353	0.268
≤10 y (N = 34)	90.7 ± 41.1	61.6 ± 35.5	−1.4	**<0.0001**		
>10 y (N = 10)	102 ± 35.1	55.2 ± 38.7	−1.9	**0.015**		
LEDD					**0.004**	0.425
≤1000 mg (N = 30)	80.9 ± 28.1	53.6 ± 28.2	−1.4	**<0.0001**		
>1000 mg (N = 14)	121.3 ± 49	68.2 ± 43.1	−1.7	**0.002**		

^a^, comparison between the score at V0 and at V6 for each group individually (Wilcoxon’s rank sum test). ^b^, comparison between patients of different group for each variable (i.e., males vs. females, etc.) in the score at V0 (Mann–Whitney–Wilcoxon test). ^c^, comparison between patients of different group for each variable (i.e., males vs. females, etc.) in the score at V6M (Mann–Whitney–Wilcoxon test). LEDD, levodopa equivalent daily dose; NMSS, Non-motor Symptoms Scale. Bold formatting means significant value (*p* < 0.005)

**Table 2 brainsci-15-00666-t002:** Change in the score of each domain of the NMSS from baseline (V0) to the final visit at 6 months (V6M) in PD patients from the SAFINOMOTOR study according to sex, age, disease duration, and LEDD.

	Males (N = 20)	Females (N = 24)	≤75 y Old (N = 33)	>75 y Old (N = 11)	≤10 y DD (N = 34)	>10 y DD (N = 10)	≤1000 mg LEDD (N = 30)	>1000 mg LEDD (N = 14)
Cardiovascular								
At V0	8.8 ± 12.1	8.7 ± 11.4	7.5 ± 11.1	12.5 ± 12.9	9.3 ± 11.1	6.7 ± 13.4	8.9 ± 11.3	8.3 ± 12.6
At V6M	8.8 ± 12.3	5 ± 9.7	3.8 ± 7.5	15.5 ± 14.9	7.3 ± 11.1	4.6 ± 10.5	7.1 ± 11.4	5.9 ± 10.2
Cohen’s d	0.0	−0.4	−0.4	0.3	−0.2	−0.2	−0.2	−0.3
*p* value	0.838	0.244	0.077	0.552	0.298	0.785	0.385	0.483
Sleep/fatigue								
At V0	34.9 ± 16	34 ± 23.7	35.3 ± 20.4	31.8 ± 21	34.9 ± 19.7	32.7 ± 23.6	31.3 ± 16.8	41.2 ± 25.9
At V6M	19.2 ± 15.2	26.5 ± 19.9	21.3 ± 16.4	28.6 ± 22.4	23.8 ± 19.4	21 ± 13.4	23.9 ± 19.5	21.6 ± 15.3
Cohen’s d	−1.6	−0.4	−1.0	−0.2	−0.7	−0.6	−0.5	−1.0
*p* value	**<0.0001**	0.253	**0.001**	0.755	**0.008**	0.126	**0.045**	**0.023**
Mood/apathy								
At V0	26.3 ± 11.5	36.3 ± 30	30.5 ± 27.8	35.3 ± 24.6	27.2 ± 26.5	47.3 ± 22.4	25 ± 22.7	46 ± 30.1
At V6M	11.5 ± 12.9	17 ± 23.8	12.7 ± 19.6	19.9 ± 19.6	14.2 ± 20.6	15.4 ± 16.9	11.3 ± 13.6	21.3 ± 28.1
Cohen’s d	−1.5	−1.2	−1.4	−1.1	−1.2	−2.2	−1.2	−1.6
*p* value	**0.001**	**<0.0001**	**<0.0001**	**0.040**	**<0.0001**	**0.005**	**<0.0001**	**0.004**
Perceptual symptoms								
At V0	3.9 ± 7.8	3.4 ± 7.3	3.7 ± 6.5	3.2 ± 9.9	4.3 ± 8.3	1.4 ± 2.6	2.7 ± 6.6	5.6 ± 8.8
At V6M	4.6 ± 7.9	1.4 ± 2.7	2.8 ± 6.1	2.8 ± 5.4	2.9 ± 5.6	2.5 ± 6.9	2.3 ± 4.9	3.9 ± 7.7
Cohen’s d	0.1	−0.4	−0.1	−0.1	−0.2	0.19	−0.1	−0.2
*p* value	0.866	0.301	0.336	0.989	0.432	0.891	0.759	0.475
Attention/memory								
At V0	20.6 ± 17.6	14 ± 17.4	16.4 ± 15.9	18.7 ± 22.7	15.2 ± 17.4	23.1 ± 17.9	13.2 ± 15.4	25 ± 19.9
At V6M	14 ± 17.1	12.7 ± 19.4	11.1 ± 12.3	19.9 ± 29.5	11.5 ± 17.3	19.4 ± 20.6	9.6 ± 15.2	21.2 ± 21.9
Cohen’s d	−0.9	−0.1	−0.7	0.1	−0.6	−0.3	−0.5	−0.3
*p* value	**0.010**	0.482	**0.006**	0.858	**0.020**	0.553	**0.032**	0.373
Gastrointestinal symptoms								
At V0	19.9 ± 17.6	17.4 ± 16.6	16.1 ± 15.8	25.8 ± 18.8	19.3 ± 17.5	15.6 ± 15.2	17.9 ± 16.1	19.6 ± 19.2
At V6M	11.9 ± 11.4	14.1 ± 14.7	10.5 ± 11.7	20.9 ± 14.9	14.6 ± 12.9	8.1 ± 13.3	11.8 ± 12	16.1 ± 15.5
Cohen’s d	−1.0	−0.3	−0.6	−0.4	−0.5	−0.5	−0.7	−0.3
*p* value	**0.019**	0.148	**0.027**	0.090	**0.042**	0.108	**0.014**	0.278
Urinary symptoms								
At V0	45 ± 29.3	42.7 ± 30.7	40.3 ± 26.3	54 ± 37.9	44.8 ± 29.6	40 ± 31.8	37.5 ± 26.4	57.1 ± 33.1
At V6M	37.2 ± 25.4	25.1 ± 21.6	29.9 ± 25.7	32.5 18.6	33.6 ± 25.2	20.6 ± 16.6	28.3 ± 20.9	35.5 ± 29.8
Cohen’s d	−0.4	−1.1	−0.6	−1.2	−0.7	−0.8	−0.6	−1.1
*p* value	0.251	**<0.0001**	**0.034**	**0.021**	**0.012**	0.086	**0.031**	**0.039**
Sexual dysfunction								
At V0	40.2 ± 36.4	19.4 ± 33.8	23.4 ± 31.8	45.4 ± 44.3	31.5 ± 37.7	20 ± 30.1	24 ± 31.1	39.3 ± 44.7
At V6M	32.1 ± 33.9	17.2 ± 32.9	20.4 ± 32.3	39.8 ± 34.4	27.6 ± 34.7	17.5 ± 29.8	18.6 ± 26.7	39.6 ± 42.5
Cohen’s d	−0.3	−0.2	−0.1	−0.2	−0.1	−0.3	−0.2	0.0
*p* value	0.672	0.812	0.823	0.888	0.875	0.565	0.639	0.799
Miscellaneous								
At V0	29.2 ± 21	34.5 ± 21.1	30.6 ± 21.1	36.3 ± 21	30.2 ± 18	38.3 ± 29.4	29.9 ± 19	36.8 ± 24.1
At V6M	16 ± 12.1	21.4 ± 15.3	19.5 ± 14.7	17.4 ± 12.3	18.7 ± 11.6	19.8 ± 21.1	18.9 ± 11.4	19.1 ± 19
Cohen’s d	−1.2	−1.1	−1.1	−1.4	−1.0	−1.6	−1.0	−1.6
*p* value	**0.002**	**0.002**	**<0.0001**	**0.013**	**<0.0001**	**0.012**	**0.001**	**0.003**

Comparison between the score at V0 and at V6 for each group individually was assessed with the Wilcoxon’s rank sum test). DD, disease duration; LEDD, levodopa equivalent daily dose; NMSS, Non-motor Symptoms Scale. The score of each domain is expressed as a percentage of the corresponding maximum possible score (e.g., 36 points in Mood/apathy is expressed as 0.5). Bold formatting means significant value (*p* < 0.005).

**Table 3 brainsci-15-00666-t003:** Change in the PDQ-39 from baseline (V0) to the final visit at 6 months (V6M) in PD patients from the SAFINOMOTOR study according to sex, age, disease duration, and LEDD.

	At V0	At V6M	Cohen’s d	*p* ^a^	*p* ^b^	*p* ^c^
Sex					0.202	0.741
Males (N = 20)	39.1 ± 23.3	32.4 ± 20.2	−0.5	0.080		
Females (N = 24)	48.3 ± 22.1	34.5 ± 22.7	−1.4	**<0.0001**		
Age					0.632	0.124
≤75 y (N = 33)	43.2 ± 22.2	30.4 ± 19.8	−1.4	**<0.0001**		
>75 y (N = 11)	46.7 ± 25.6	42.4 ± 24	−0.3	0.306		
Disease duration					**0.009**	0.105
≤10 y (N = 34)	40.3 ± 23.5	31.2 ± 21.5	−0.8	**0.002**		
>10 y (N = 10)	57.9 ± 14.2	42.1 ± 19.9	−2.1	**0.011**		
LEDD					0.391	0.265
≤1000 mg (N = 30)	40.9.9 ± 18.5	29.1 ± 17.1	−1.1	**<0.0001**		
>1000 mg (N = 14)	50.6 ± 31.8	41.9 ± 26.8	−0.6	0.119		

^a^, comparison between the score at V0 and at V6 for each group individually (Wilcoxon’s rank sum test). ^b^, comparison between patients of different group for each variable (i.e., males vs. females, etc.) in the score at V0 (Mann–Whitney–Wilcoxon test). ^c^, comparison between patients of different group for each variable (i.e., males vs. females, etc.) in the score at V6M (Mann–Whitney–Wilcoxon test). LEDD, levodopa equivalent daily dose; PDQ-39, 39-item Parkinson’s Disease Quality of Life Questionnaire. Bold formatting means significant value (*p* < 0.005).

**Table 4 brainsci-15-00666-t004:** Change in the score of each domain of the PDQ-39 from baseline (V0) to the final visit at 6 months (V6M) in PD patients from the SAFINOMOTOR study according to sex, age, disease duration, and LEDD.

	Males (N = 20)	Females (N = 24)	≤75 y Old (N = 33)	>75 y Old (N = 11)	≤10 y DD (N = 34)	>10 y DD (N = 10)	≤1000 mg LEDD (N = 30)	>1000 mg LEDD (N = 14)
Mobility								
At V0	22.3 ± 30.3	42.5 ± 27.6	31.4 ± 25.9	38.9 ± 28.2	28.3 ± 25.7	50.3 ± 21.9	30.5 ± 20.7	39.3 ± 35.7
At V6M	23.5 ± 22.8	33.8 ± 29.4	23.5 ± 22.7	45.9 ± 32.1	25.9 ± 25.6	40 ± 29.5	24.3 ± 20.4	39.2 ± 35.9
Cohen’s d	0.1	−0.6	−0.7	0.5	−0.2	−0.9	−0.5	0.0
*p* value	0.584	**0.017**	**0.004**	0.592	0.184	0.074	**0.021**	0.574
Activities of daily living								
At V0	24.6 ± 23.4	27.6 ± 22.1	25.3 ± 21.9	29.2 ± 24.9	22.4 ± 20.2	39.2 ± 25.9	23.1 ± 21.8	33 ± 23.2
At V6M	18.3 ± 18.1	17.4 ± 18.2	14.9 ± 15.6	26.5 ± 22.2	14.8 ± 15.4	27.9 ± 22.9	15.2 ± 15.3	23.2 ± 22.3
Cohen’s d	−0.5	−0.8	−0.9	−0.2	−0.6	−0.8	−0.6	−0.8
*p* value	0.244	**0.019**	**0.002**	0.964	**0.047**	0.114	0.119	0.073
Emotional well-being								
At V0	31.4 ± 23.4	48.4 ± 27	40.4 ± 26.9	42.4 ± 27.1	38.6 ± 26.6	48.3 ± 26.7	39.4 ± 24.2	44.2 ± 32.3
At V6M	22.1 ± 20.9	29.8 ± 24.4	24.6 ± 24.1	31.4 ± 19.6	26.6 ± 23.3	25.4 ± 23	23.1 ± 20.2	33.3 ± 27.7
Cohen’s d	−0.8	−1.3	−1.1	−1.0	−0.9	−1.9	−1.1	−1.1
*p* value	**0.040**	**<0.0001**	**<0.0001**	0.052	**0.002**	**0.008**	**<0.0001**	**0.022**
Stigmatization								
At V0	21.7 ± 25.4	9.1 ± 10.9	13.9 ± 18.8	17 ± 22.4	14.4 ± 18.9	15.6 ± 22.4	15.4 ± 18.7	12.9 ± 21.9
At V6M	11.9 ± 16.1	4.2 ± 8.9	6.8 ± 12	10.2 ± 16.3	6.4 ± 12.3	11.8 ± 15.4	9.6 ± 14.4	3.6 ± 8.7
Cohen’s d	−0.6	−0.5	−0.5	−0.9	−0.6	−0.3	−0.6	−0.5
*p* value	0.070	0.107	0.057	0.063	**0.011**	0.492	**0.024**	0.207
Social support								
At V0	7.5 ± 8.7	4.5 ± 10.7	5.2 ± 9.1	7.6 ± 12	4.5 ± 8.9	10 ± 12.3	5 ± 9.7	7.7 ± 10.5
At V6M	3.3 ± 11.3	3.8 ± 193.9	4.3 ± 14.4	1.5 ± 3.8	2.2 ± 8.8	8.3 ± 21.1	4.4 ± 14.9	1.8 ± 4.8
Cohen’s d	−0.4	−0.1	−0.1	−0.7	−0.3	−0.2	−0.1	−0.7
*p* value	0.121	0.752	0.651	0.131	0.181	0.750	0.643	0.111
Cognition								
At V0	27.9 ± 20.9	22.1 ± 16.9	26.2 ± 19.7	20.4 ± 16.1	25.4 ± 20.5	22.5 ± 12.6	23.1 ± 15.1	28.4 ± 25.9
At V6M	28.4 ± 25.3	19.8 ± 19.5	24.4 ± 22	21.6 ± 24.9	22.9 ± 21.6	26.2 ± 26.3	19.4 ± 19	33 ± 27.1
Cohen’s d	0.0	−0.2	−0.1	0.1	−0.2	0.2	−0.3	0.3
*p* value	0.909	0.158	0.165	0.893	0.295	0.918	0.193	0.929
Communication								
At V0	25.9 ± 27.1	7.6 ± 10.9	15.6 ± 22.4	15.9 ± 19.9	15.9 ± 21.7	15 ± 25.4	14.7 ± 21.5	17.9 ± 22.3
At V6M	18.2 ± 17.5	6.3 ± 9.9	12.4 ± 15	11.4 16.3	12.7 ± 15.8	10 ± 13.9	9.2 ± 11.8	18.4 ± 19.6
Cohen’s d	−0.3	−0.2	−0.2	−0.3	−0.2	−0.5	−0.4	0.1
*p* value	0.278	0.497	0.399	0.344	0.367	0.380	0.090	0.980
Pain and discomfort								
At V0	37.2 ± 26.6	48.3 ± 27.7	43.5 ± 27.2	43.2 ± 29.3	44.2 ± 26.9	40.8 ± 30.3	42.8 ± 22.8	44.8 ± 37
At V6M	27.9 ± 20.9	37.8 ± 18.2	34.8 ± 19.4	28.8 ± 21.5	33.3 ± 19.8	33.3 ± 21.1	32.5 ± 19.6	35.1 ± 21.2
Cohen’s d	−0.6	−0.5	−0.5	−1.0	−0.6	−0.3	−0.7	−0.4
*p* value	0.070	0.080	0.076	0.043	**0.019**	0.362	**0.016**	0.307

Comparison between the score at V0 and at V6 for each group individually was assessed with the Wilcoxon’s rank sum test. DD, disease duration; LEDD, levodopa equivalent daily dose; The score of each domain is expressed as a percentage of the corresponding maximum possible score (e.g., 20 points in Mobility is expressed as 0.5). Bold formatting means significant value (*p* < 0.005).

## Data Availability

The protocol and statistical analysis plan are available upon request.

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
