# Peer review of "Effect of Safinamide on Non-Motor Symptoms and Quality of Life in Parkinson’s Disease Patients According to Sex, Age, Disease Duration and Levodopa Equivalent Daily Dose"

_brainsci, 2025, doi:10.3390/brainsci15070666_

Round 1

Reviewer 1 Report

Comments and Suggestions for Authors
  1. The Abstract needs improvement by revising the parts of background/objective where actual brief introduction of the hypothesis presentation would be better rather putting the result and analysis. Further, conclusion part here can a bit more conclusive.

  1. Same goes for the keywords. More appropriate keywords can be kept which suits more to the manuscript. For example, why should you have “Effectiveness” as a keyword?

  1. Authors mentioned that “It has been approved in various 58 regions, including Europe, the United States, Asia, and Japan, for the treatment of motor 59 fluctuations in PD”. Please correct this sentence as Japan itself is Asia. Moreover, will you please clarify that if it is US-FDA approved? If yes, then is it only in those 58 regions or globally?

  1. In the title you mentioned Safinamide is for improving non-motor symptoms of PD. Conversely, you also mentioned that “Specifically, safinamide has demonstrated benefits for motor symptoms such as tremor, bradykinesia, rigidity, and gait” in the introduction. Please correct the sentence or provide justifications. Is it specifically for motor or non-motor symptoms. Has it been approved for motor symptoms or non-motor symptoms?

  1. Safinamide 50 mg tablet should be provided with brand name. Was the dose adjusted for different age groups, body weight etc.?

  1. Exclusion and inclusion criteria should clearly be mentioned.

  1. What do you mean by urinary symptoms? Please explain so that it will help the readers. Moreover, the type and area of pain can also be explained.

Author Response

- The Abstract needs improvement by revising the parts of background/objective where actual brief introduction of the hypothesis presentation would be better rather putting the result and analysis. Further, conclusion part here can a bit more conclusive.

We appreciate your suggestion. We acknowledge the importance of presenting a clear and structured Abstract. However, we would like to point out that the current version follows the typical format of abstracts in clinical research, including background, methods, key findings, and conclusion. Moreover, this abstract was previously presented and published in the proceedings of a scientific congress. Nevertheless, we understand the reviewer’s perspective and are open to revising the Abstract if the editor considers it necessary for consistency with the journal’s format and expectations.

- Same goes for the keywords. More appropriate keywords can be kept which suits more to the manuscript. For example, why should you have “Effectiveness” as a keyword?

Thank you for this observation. We have revised the keywords to better reflect the scope and focus of the manuscript. The keyword “Effectiveness” has been replaced with more specific and relevant terms. Please refer to the revised keyword list in the updated manuscript.

- Authors mentioned that “It has been approved in various 58 regions, including Europe, the United States, Asia, and Japan, for the treatment of motor 59 fluctuations in PD”. Please correct this sentence as Japan itself is Asia. Moreover, will you please clarify that if it is US-FDA approved? If yes, then is it only in those 58 regions or globally?

Many thanks for this observation. While Japan is geographically part of Asia, we initially mentioned it separately to reflect its distinct regulatory framework (PMDA), which is independent from those of other Asian countries. However, to avoid any potential confusion, we have rephrased the sentence in the manuscript as follows:

“It has been approved in various regions, including Europe, the United States, Japan, and other parts of Asia, for the treatment of motor fluctuations in PD.”

We believe this revised wording maintains the intended meaning while improving clarity and consistency.

- In the title you mentioned Safinamide is for improving non-motor symptoms of PD. Conversely, you also mentioned that “Specifically, safinamide has demonstrated benefits for motor symptoms such as tremor, bradykinesia, rigidity, and gait” in the introduction. Please correct the sentence or provide justifications. Is it specifically for motor or non-motor symptoms. Has it been approved for motor symptoms or non-motor symptoms?

Thank you very much for this comment. We would like to clarify that there is no contradiction between the title and the content of the manuscript. Safinamide has indeed been approved for motor symptoms in Parkinson’s disease. This is clearly stated in the introduction to provide context. The novelty and focus of this manuscript lies in its evaluation of the drug’s additional impact on non-motor symptoms (NMS), as demonstrated in recent studies including SAFINONMOTOR. Therefore, the statement in the introduction is intended to reflect the full therapeutic profile of the drug. No changes have been made in this section, as we believe the current text is accurate and appropriately contextualized.

- Safinamide 50 mg tablet should be provided with brand name. Was the dose adjusted for different age groups, body weight etc.?

We thank the reviewer for this comment. We do not use the brand name (Xadago®) because we understand that it is not appropriate in a scientific publication. As explained in the manuscript, the following indication existed in the SAFINONMOTOR study protocol: “Safinamide was administered as a 50 mg tablet once daily for one month and increased to 100 mg/day at V2. In some cases (e.g., dyskinesia), the 100 mg dose could be introduced earlier, or the dose could be maintained at 50 mg/day based on the neurologist's judgment”. Therefore, based on this premise, the dose could be adjusted at the researcher's discretion. What you mention about weight adjustment is very interesting, but unfortunately, it was not specifically considered.

- Exclusion and inclusion criteria should clearly be mentioned.

Thank you again for the comment. As this manuscript presents a secondary analysis based on the SAFINONMOTOR study, we have not repeated the inclusion and exclusion criteria here. However, the manuscript explicitly states that the materials and methods, including patient selection criteria, are thoroughly described in the original SAFINONMOTOR publication. A reference to that article is provided for full methodological details. We believe this approach avoids redundancy while ensuring transparency and traceability.

- What do you mean by urinary symptoms? Please explain so that it will help the readers. Moreover, the type and area of pain can also be explained.

Many thanks for the comment. The urinary symptoms referred to in the manuscript are included as part of the Non-Motor Symptoms Scale (NMSS), a validated and widely used tool that comprehensively captures various non-motor symptoms in Parkinson’s disease. As these symptoms are already systematically assessed and described within the NMSS - "Urinary symptoms (items 22, 23 and 24; score, 0 to 36) -, we believe that an additional detailed explanation in the manuscript is not necessary. Therefore, no changes have been made regarding this point.

Reviewer 2 Report

Comments and Suggestions for Authors

In this study, the authors find that safinamide treatment over six months significantly improved non-motor symptoms in Parkinson´s Disease patients independently of sex, age, DD and LEDD. These results preliminarily reinforce the clinical utility of safinamide as an adjunctive therapy in Parkinson´s Disease. This work is expected to attract broad interest among researchers in the fields of biomedical science. Therefore, I recommend the publication of this manuscript on brain sciences after minor revisions. Specific suggestions are as follows:

1. In Figure1 and Figure 2 captions, authors claim that Mild outliers (O) are data points that are more extreme than Q1-1.5. I am curious that whether it should be Q3+1.5IQR?

2. Since this study included 4 scheduled visits from V0 to V6M. Some figures showing the change in patient scores over time points may be helpful in demonstrating the effect of the safinamide.

Author Response

In this study, the authors find that safinamide treatment over six months significantly improved non-motor symptoms in Parkinson´s Disease patients independently of sex, age, DD and LEDD. These results preliminarily reinforce the clinical utility of safinamide as an adjunctive therapy in Parkinson´s Disease. This work is expected to attract broad interest among researchers in the fields of biomedical science. Therefore, I recommend the publication of this manuscript on brain sciences after minor revisions. Specific suggestions are as follows:

  1. In Figure1 and Figure 2 captions, authors claim that Mild outliers (O) are data points that are more extreme than Q1-1.5. I am curious that whether it should be Q3+1.5IQR?

Many thanks for the comment. You are absolutely right. It is be Q3+1.5IQR. It has been corrected according to comment.

  1. Since this study included 4 scheduled visits from V0 to V6M. Some figures showing the change in patient scores over time points may be helpful in demonstrating the effect of the safinamide.

Many thanks again for your observation. Figure 1 and Figure 2 have been modified according to it. The legend of the figures have been also changed:

Figure 1. Change in the NMSS total score from baseline (V0) to the final visit at 6 months (V6M) in PD patients from the SAFINOMOTOR study according to sex, age, disease duration and LEDD (data of all visits are shown). Data are presented as box plots, with the box representing the median and the two middle quartiles (25-75%). Mild outliers (O) are data points that are more extreme than Q1 - 1.5 (case number shown). LEDD, levodopa equivalent daily dose; NMSS, Non-motor Symptoms Scale.

Figure 2. Change in the PDQ39 total score from baseline (V0) to the final visit at 6 months (V6M) in PD patients from the SAFINOMOTOR study according to sex, age, disease duration and LEDD ((data of all visits are shown). Data are presented as box plots, with the box representing the median and the two middle quartiles (25-75%). Mild outliers (O) are data points that are more extreme than Q1 - 1.5 (case number shown). LEDD, levodopa equivalent daily dose; NMSS, PDQ-39, 39-item Parkinson’s Disease Quality of Life Questionnaire.

Furthermore, we observed that some errors had been made in the previous figures, given that the variable's data corresponded to another visit that had been included by mistake. For example, in the LEDD analysis, the V3M data was included in the >1,000 mg/day group instead of the V6M, by mistake. We confirm that the figures with data from all visits are correct. Many thanks again.

Reviewer 3 Report

Comments and Suggestions for Authors

Safinamide (((S)-(+)-2-(4-(3-fluorobenzyloxy)benzylamino)propanamide)) is being developed as a treatment for PD. It combines several mechanisms of action that can be beneficial for this disease, including potentiation of dopaminergic neurotransmission via monoamine oxidase B (MAO-B), dopamine reuptake inhibition, and modulation of glutamatergic neurotransmission with reduction in oxidative damage, which may provide neuroprotective action and improvement of cognitive function. The authors presented a well-structured investigatation on the impact of analysis of the SAFINONMOTOR study the effect of safinamide on wide range of non-motor symptoms burden and QoL in order to provide further evidence of its clinical impact and potential applications in personalized treatment for PD. This study is a continuation of the previously published results from the SAFINONMOTOR Study (Pain Improvement in Parkinson's Disease Patients Treated with Safinamide: [Refs. 22-24]. As a positive aspect of the study conducted in this work, it should be noted the breadth of the experimental design both in terms of age and the degree of disease progression. As a results, in this post hoc analysis, the significant improvement detected in the non-motor symptoms burden was present across all groups, including in those patients with more clinically complex profiles, such as patients over 75 years old, those with a disease exceeding  10 years, and patients receiving a high LEDD (>1000 mg/day). As for QoL, gains were noted across the board but reached statistical significance particularly in women, patients no older than 75 years old, and those receiving a LEDD no higher than 1000 mg, regardless of disease duration (≤ 10 years or >10 years).These findings highlight the potential of safinamide  as an adjunct therapy for NMS across various patient subgroups and stages of progression.

Recommendations for Revision and Improvement

 The main questions to the presented manuscript are related to the study design itself

- why the study didn't use a placebo group and there is not a comparative  with placebo

- In certain subgroups the sample size was relatively small, particularly (Table 2 and Table 4, > 75 y old (N=11), which may have limited the statistical power to detect significant differences.

- Including of patients with a severe or very severe non-motor symptoms burden (DD>10 n 10) also have limited the statistical power and reduce the generalizability of the results to all patients with PD.

The potential answers and limited additions would strengthen the context of the manuscript.

By addressing these points, the study will achieve a higher level of clarity, further supporting its publication.

Author Response

The main questions to the presented manuscript are related to the study design itself

- why the study didn't use a placebo group and there is not a comparative with placebo

Thank you very much for your observation. Indeed, the absence of a placebo group is a recognized limitation of the study, which we have explicitly acknowledged in the manuscript. This observational study was designed as a post hoc analysis based on real-world data from the SAFINONMOTOR study, where all patients received safinamide as part of routine clinical care. Therefore, a placebo-controlled design was not feasible in this context. We have emphasized this point in the limitations section to ensure transparency

- In certain subgroups the sample size was relatively small, particularly (Table 2 and Table 4, > 75 y old (N=11), which may have limited the statistical power to detect significant differences.

Many thanks for highlighting this important point. The sample size for some subgroups—specifically the >75 years old group (N=11)—is indeed small and may have influenced the statistical outcomes. However, this limitation has been carefully considered and discussed in the manuscript. It is explicitly mentioned that the small sample size in certain subgroups might have affected the results. Furthermore, in the limitations section, the manuscript states:

"Second, the sample size was relatively small, particularly in certain subgroups, which may have limited the statistical power to detect significant differences."

Additionally, the conclusions emphasize the need for further studies with larger sample sizes to confirm and extend these findings:

"Further studies with larger sample sizes are needed to validate these results and explore more thoroughly how treatment response varies across demographic and clinical factors in order to optimize safinamide’s clinical use and maximize its therapeutic benefits for non-motor symptoms in PD patients.”

- Including of patients with a severe or very severe non-motor symptoms burden (DD>10 n 10) also have limited the statistical power and reduce the generalizability of the results to all patients with PD.

Thank you very much again for this valuable comment. We acknowledge that the inclusion of patients with a severe or very severe non-motor symptoms burden, particularly those with disease duration greater than 10 years (N=10), may limit the statistical power and reduce the generalizability of the results. This limitation is explicitly stated and discussed in the manuscript as follows:

"Third, the inclusion of patients with a severe or very severe NMS burden may reduce the generalizability of the results to all patients with PD."

We agree that further studies with larger and more diverse patient populations are necessary to validate and extend these findings."
